# Treatment sequences of patients with advanced colorectal cancer and use of second-line FOLFIRI with antiangiogenic drugs in Japan: A retrospective observational study using an administrative database

**Eiji Shinozaki[1], Akitaka Makiyama[2], Yoshinori Kagawa[3], Hironaga Satake[4], Yoshinori Tanizawa[5]\*, Zhihong Cai[5], Yongzhe Piao[5]**

**1** Gastroenterology Center, Japanese Foundation for Cancer Research, Cancer Institute Hospital, Tokyo, Japan, **2** Cancer Center, Gifu University Hospital, Gifu, Japan, **3** Department of Gastroenterological Surgery, Osaka General Medical Center, Osaka, Japan, **4** Cancer Treatment Center, Kansai Medical University Hospital, Osaka, Japan, **5** Medicines Development Unit-Japan, Eli Lilly Japan K.K., Kobe, Japan

\* tanizawa_yoshinori@lilly.com

**Data Availability Statement:** The data in this study belong to Medical Data Vision Co., Ltd (http://www.

## Abstract

The objectives were to describe treatment sequences for advanced colorectal cancer (CRC), use of second-line FOLFIRI (leucovorin, 5-fluorouracil, irinotecan) plus antiangiogenic drug (bevacizumab, ramucirumab, aflibercept beta) therapy, and the factors associated with the duration of antitumor drug treatment from second-line antiangiogenic therapy in Japan. This retrospective observational study was conducted using a Japanese hospital-based administrative database. Patients were enrolled if they started adjuvant therapy (and presumably experienced early recurrence) or first-line treatment for advanced CRC between May 2016 and July 2019, and were analysed until September 2019. Factors associated with overall treatment duration from second-line treatment with FOLFIRI plus antiangiogenic drugs were explored with multivariate Cox regression analysis. The most common first-line treatments were FOLFOX (leucovorin, 5-fluorouracil, oxaliplatin) or CAPOX (capecitabine, oxaliplatin) with bevacizumab (presumed *RAS*-mutant CRC) and FOLFOX with panitumumab (presumed *RAS*-wild type CRC). The most common second-line treatments were FOLFIRI-based. Many patients did not transition to subsequent lines of therapy. For second-line treatment, antiangiogenic drugs were prescribed more often for patients with presumed *RAS*-mutant CRC, right-sided CRC, and independent activities of daily living (ADL). The median duration of second-line FOLFIRI plus antiangiogenic drug treatment was 4.5 months; 66.2% of patients transitioned to third-line therapy. Low body mass index and not fully independent ADL were significantly associated with shorter overall duration of antitumor drug treatment from second-line therapy. Left-sided CRC, presumed *RAS*-wild type CRC, previous use of oral fluoropyrimidines and use of proteinuria qualitative tests, antihypertensives, or anticholinergics during second-line therapy were significantly associated with longer treatment. Treatment of advanced CRC in Japan is consistent with both international and Japanese guidelines, but transition rates to subsequent therapies need

mdv.co.jp/) and were used under licence, funded by Eli Lilly Japan K.K. As the data are not publicly available, researchers looking to access the data used in this study should contact Medical Data Vision Co., Ltd via their website (http://www.mdv. co.jp/ [Japanese] or https://en.mdv.co.jp/ [English]).

**Funding:** This study was sponsored by Eli Lilly Japan K.K., manufacturer/licensee of ramucirumab. Medical writing assistance provided by Serina Stretton, PhD, CMPP, and Rebecca Lew, PhD, CMPP, of ProScribe – Envision Pharma Group was funded by Eli Lilly Japan K.K. The funder provided support in the form of salaries for authors (YT, ZC, and YP) and was involved in the study design, data collection, data analysis, preparation of the manuscript, and in the decision to publish.

**Competing interests:** ES has received honoraria from Taiho Pharmaceutical Co., Ltd., Takeda Pharmaceutical Co., Ltd., Chugai Pharmaceutical Co., Ltd., Eli Lilly Japan Co., Ltd., Merck Bio Pharma Co., Ltd., Sanofi Co., Ltd., and Yakult Honsha Co., Ltd. YK has received honoraria from Bayer Co., Ltd., Chugai Pharmaceutical Co., Ltd., Yakult Honsha Co., Ltd., Sanofi Co., Ltd., Eli Lilly Japan Co., Ltd., Taiho Pharmaceutical Co., Ltd., Takeda Pharmaceutical Co., Ltd., and Merck Co., Ltd. AM has received honoraria from Eli Lilly Japan Co., Ltd., Chugai Pharmaceutical Co., Ltd., and Takeda Pharmaceutical Co., Ltd. HS has received research funding from Ono Pharmaceutical Co., Ltd., Taiho Pharmaceutical Co., Ltd., and Takeda Pharmaceutical Co., Ltd., and honoraria from Bayer Co., Ltd., Bristol-Myers Squibb Co., Ltd., Chugai Pharmaceutical Co., Ltd., Daiichi Sankyo Co., Ltd., Eli Lilly Japan Co., Ltd., Merck Bio Pharma Co., Ltd., MSD Co., Ltd., Ono Pharmaceutical Co., Ltd., Sanofi Co., Ltd., Taiho Pharmaceutical Co., Ltd., Takeda Co., Ltd., and Yakult Honsha Co., Ltd. This study was sponsored by Eli Lilly Japan K.K., manufacturer/licensee of ramucirumab. YT, ZC, and YP are employees and/or minor shareholders of Eli Lilly Japan K.K. Medical writing assistance provided by Serina Stretton, PhD, CMPP, and Rebecca Lew, PhD, CMPP, of ProScribe – Envision Pharma Group was funded by Eli Lilly Japan K.K. This does not alter our adherence to PLOS ONE policies on sharing data and materials, but the data used for this study are owned by a third party (Medical Data Vision Co. Ltd.).

improvement. In addition to antitumor drug treatment, better ADL, higher body mass index, management of hypertension, and proteinuria tests were associated with continuation of sequential therapy that included antiangiogenic drugs.

## Introduction

Colorectal cancer (CRC) is the third most common cancer and the second most common cause of cancer-related deaths worldwide [1]. In Japan in 2016, 158,127 people were diagnosed with CRC [2], and in 2018 there were 50,658 CRC deaths [2]. The number of new patients with CRC and the number of deaths attributed to CRC in Japan have continued to increase, despite the availability of new treatments and treatment modalities for CRC [2–4].

The Japanese Society for Cancer of the Colon and Rectum (JSCCR) guidelines for treatment of CRC recommend three antiangiogenic drugs as second-line treatment for unresectable advanced or recurrent CRC [4]. Bevacizumab is recommended in combination with various chemotherapy regimens including FOLFIRI (leucovorin, 5-fluorouracil [5-FU], irinotecan). Two more antiangiogenic drugs, ramucirumab and aflibercept beta, are recommended only as FOLFIRI combination therapy; therefore, there are three available antiangiogenic drugs that can be chosen in combination with FOLFIRI for second-line treatment. Bevacizumab was the first to become available for clinical use in Japan in April 2007. More recently, ramucirumab was approved for second-line use (May 2016) based on the results of the international phase III RAISE trial [5], and aflibercept beta was approved for second-line use (March 2017) based on the results of a Japanese phase II trial [6] and the phase III VELOUR trial, which did not include Japanese patients [7].

Although the clinical trials for ramucirumab [5] and aflibercept beta [6] included Japanese patients, there is limited information on the real-world use of these new drugs in patients with CRC in Japan. Moreover, since these trials were conducted, new treatment options have become available and the recommended treatment sequences have been updated [4]. Therefore, in the pursuit of optimal treatment options for patients with advanced CRC, it is important to understand the current treatment sequences and how these three antiangiogenic drugs are used in current real-world clinical practice in Japan.

Because administrative databases capture the daily medical activities conducted in clinical practice, they are increasingly being used as a data source to investigate real-world treatment patterns [8]. However, a disadvantage of these databases is a lack of clinical and disease-related information. Once approved, drugs are prescribed to individual patients based on their clinician's medical advice, with the choice of drug depending on the patient's status. This can introduce confounding by indication [9, 10] when attempting to compare the effectiveness of different drugs. This is particularly relevant when using administrative databases because many of the potentially relevant factors needed for background adjustment are not available [11, 12]. Nevertheless, administrative databases are valuable for understanding how drugs are used in the real-world and for generating hypotheses to explore in future.

The primary objective of this study was to describe the treatment sequences of patients with advanced CRC in Japan using an administrative database. The secondary objectives were to describe the use of antiangiogenic drugs in combination with FOLFIRI as second-line treatment, and to explore the clinical and treatment-related factors associated with overall treatment duration from the second-line antiangiogenic therapies.

## Materials and methods

### Study design

This was a retrospective observational cohort study that used the Medical Data Vision Co. Ltd. (MDV) database (Tokyo, Japan) [13]. MDV is a large hospital-based database in Japan [13] that includes administrative claims and Diagnosis Procedure Combination (DPC) data from hospitalisations and outpatient visits of patients who attend hospitals participating in the DPC system (DPC hospitals). DPC is a per-day flat-sum payment system for acute inpatient care [14, 15]. As such, DPC hospitals provide acute-phase medical care but may also provide other medical care. As of December 2019, the MDV database included approximately 29.5 million patients from 393 DPC hospitals in Japan (i.e., approximately 23% of DPC hospitals). The ethical principles under which the study was conducted were in accordance with the Declaration of Helsinki and were consistent with Good Pharmacoepidemiology Practices. Consistent with the Japanese Ethical Guidelines for Medical and Health Research Involving Human Subjects, ethical review and informed consent were not required because the study used retrospective de-identified data.

### Study populations

The antitumor drugs included in the analyses were those listed in the JSCCR guidelines as options for systemic therapy for advanced CRC [4]: cytotoxic agents included fluoropyrimidine (capecitabine, 5-FU, combination tegafur/gimestat/potassium otastat [S-1], combination uracil/tegafur [UFT]), combination trifluridine/tipiracil (FTD/TPI), irinotecan, and oxaliplatin; antiangiogenic drugs included aflibercept beta, bevacizumab, and ramucirumab; anti-epidermal growth factor receptor (EGFR) antibody drugs included cetuximab and panitumumab; and other targeted treatments included pembrolizumab and regorafenib.

Patients for this study were derived from the population in the MDV database with at least one confirmed neoplasm diagnosis (*International Statistical Classification of Diseases and Related Health Problems*, *10th Revision* [ICD-10] codes C00–D48) between April 2008 and September 2019 (data cutoff by MDV on November 25th, 2019). From this starting population, three populations were included in this analysis: a first-line population, an early recurrence population, and a FOLFIRI plus antiangiogenic drug subpopulation (Fig 1). The eligibility criteria for all populations were patients with a record of a confirmed diagnosis of CRC (*ICD-10* codes C18–20) between May 2014 and July 2019 and age ≥20 years at the time of the first prescription for first-line therapy (first-line population) or first colorectal resection surgery (early recurrence population) after the first CRC diagnosis. Patients who were prescribed antitumor drugs not recommended in the JSCCR guidelines as systemic therapy for CRC [4] were excluded.

Patients in the first-line population were presumed to have advanced or metastatic disease. Patients were eligible for inclusion in this population if they had a prescription for monoclonal antibodies recommended for use as first-line therapy for advanced disease in the JSCCR guidelines (i.e., bevacizumab, cetuximab, panitumumab) [4] after the first CRC diagnosis between May 2014 and July 2019. Patients were excluded if they had previously been prescribed bevacizumab, cetuximab, or panitumumab for 1 year before this first prescription or if they were eligible for the early recurrence population.

Patients in the early recurrence population were presumed to have had early recurrence of their cancer after resection surgery and adjuvant therapy. Patients were eligible for this population if they had undergone colorectal resection surgery after the first CRC diagnosis between May 2014 and July 2019, started adjuvant therapy (i.e., capecitabine, 5-FU, S-1, UFT, or

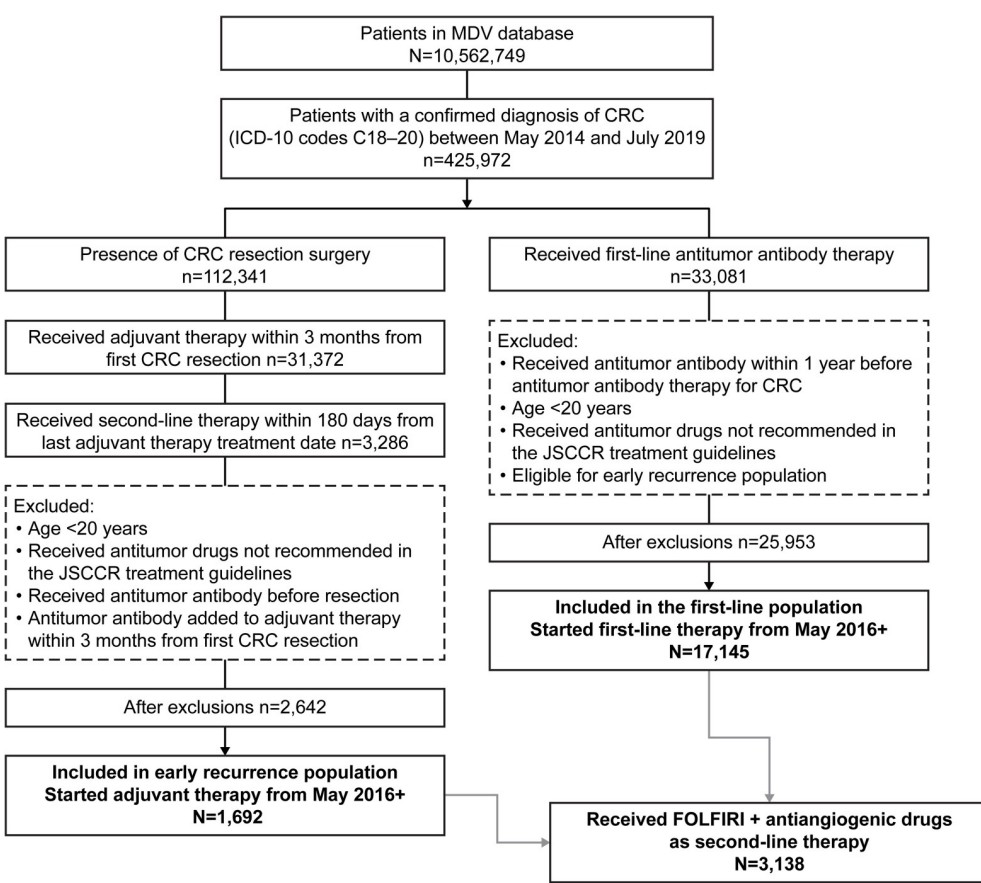

**Fig 1. Patient flow diagram.** MDV, Medical Data Vision; ICD-10, *International Statistical Classification of Diseases and Related Health Problems*, *10th Revision*; CRC, colorectal cancer; JSCCR, Japanese Society for Cancer of the Colon and Rectum; FOLFIRI, leucovorin, 5-fluorouracil, and irinotecan.

oxaliplatin) within 3 months after the surgery, and switched to second-line therapy within 6 months from the last dose of adjuvant therapy (presumably owing to a recurrence). Patients were excluded if they were prescribed bevacizumab, cetuximab, or panitumumab before the first colorectal resection surgery or had an added prescription for bevacizumab, cetuximab, or panitumumab within 3 months from the start of chemotherapy (because such treatment may not have been classified as adjuvant therapy).

Patients who started first-line/adjuvant therapy in or after May 2016 (when more than one antiangiogenic drug was available for second-line therapy in Japan) were included in the analysis (Fig 1). The index date was the date of the first prescription for second-line therapy, if present.

Patients were included in the FOLFIRI plus antiangiogenic drug subpopulation if they had met the criteria for the first-line or early recurrence populations and had received combination FOLFIRI plus one of the antiangiogenic drugs (bevacizumab, ramucirumab, or aflibercept beta) as second-line therapy (Fig 1).

## Baseline characteristics

Patient baseline characteristics were analysed in the baseline period, which started 1 year before the index date and ended on the index date. Because rat sarcoma viral oncogene

homolog (*RAS*) mutation status is a critical determinant of treatment [4, 16] and was not available in the MDV database, patients were grouped into those who were and were not prescribed anti-EGFR monoclonal antibodies (cetuximab or panitumumab) in the whole analysis period (May 2014–September 2019). Patients with a prescription for anti-EGFR antibody were presumed to have *RAS*-wild type CRC because these antibodies are only recommended for patients with *RAS*-wild type metastatic CRC [4]. Patients without a prescription for anti-EGFR antibody were presumed to have *RAS*-mutant CRC.

## Treatment sequences

For the first-line population, first-line therapy started at the first prescription of bevacizumab, cetuximab, or panitumumab. For the early recurrence population, adjuvant therapy started at the first prescription for capecitabine, 5-FU, S-1, UFT, or oxaliplatin within 3 months after surgery. First-line and adjuvant therapies were defined as the combination of all drugs prescribed within the initial 28-day period of the first prescription. The end of first-line or adjuvant therapy was defined either as 1) discontinuation of all antitumor drugs for 180 days or more, or 2) prescription of new antitumor drugs that were not included in the first-line or adjuvant treatment regimen, whichever occurred first. Second and later lines of therapy started at the first prescription for antitumor drugs after the previous line of therapy ended. The end of second and later lines of therapy was defined in the same way as for first-line or adjuvant therapy, except that addition of biologics to treatment regimens that did not include biologics (i.e., chemotherapy-only regimens) was regarded as an addition to the treatment regimen and not termination of the therapy line. Addition of biologics to first-line or adjuvant therapy was not applicable.

Transition rates between lines of therapy were evaluated in patients who were likely to have completed their previous line of therapy. The lines of therapy were considered to have been completed when the interval between the end date of each line of therapy and the end of available hospital data was more than 60 days or a subsequent line of therapy existed. Patients were considered to have completed all antitumor systemic therapies when the time between the expected final dose date of any antitumor drugs and the final date of the data provided from the hospital was 60 days or greater. Otherwise, the treatment was censored at the expected final dose date.

## Statistical analysis

Descriptive statistics of collected data are presented as n (%), mean (standard deviation), and/or median (interquartile range), as applicable, for patients in each group or cohort.

Analysis of the factors associated with the number of antiangiogenic drug prescriptions was conducted using a multivariate logistic regression model. The dependent variable was the number of antiangiogenic drug prescriptions during second-line therapy (1 vs ≥2), and the covariates were as follows: designated cancer hospital; age ≥70 years at the start of second-line therapy; sex; left-sided CRC; presence of anti-EGFR antibody prescription (presumed *RAS*-wild type CRC), body mass index (BMI) ≤18.5 kg/m$^2$; any activities of daily living (ADL)–not independent at baseline (assessed using the 10-item Barthel ADL index that measures functional independence [17]); treatment with oral fluoropyrimidine, or irinotecan in the previous line of therapy; duration of the previous line of therapy ≥180 days; and early recurrence.

Analysis of the factors associated with overall treatment duration from the start of second-line treatment with FOLFIRI plus antiangiogenic drugs was conducted using a multivariate Cox regression model. The dependent variable was overall treatment duration from the start of second-line therapy with antiangiogenic drugs to the end of all antitumor drug therapies.

The covariates included those listed in the logistic regression above and concomitant quantitative and qualitative proteinuria tests and prescription of antihypertensives, anticholinergics, and anticoagulants during second-line therapy.

The Kaplan-Meier method was used to estimate the median and 95% confidence interval (CI) for the duration of second-line antiangiogenic drug treatment.

The Instant Health Data platform (Panalgo, Boston, MA, USA) was used for cohort selection and creation of analytic variables, and all statistical analyses were conducted using R version 3.2.1 (R Foundation for Statistical Computing, Vienna, Austria) or SAS version 9.4 (SAS Institute, Cary, NC, USA).

## Results

### Patient selection

During the patient selection period, a confirmed diagnosis of CRC was recorded in 425,972 patients (Fig 1). Of these, 1,692 met the criteria for the early recurrence population and received adjuvant therapy in or after May 2016, and 17,145 met the criteria for the first-line population and received first-line antitumor therapy in or after May 2016. Within the combined first-line and early recurrence populations, 3,138 patients (1,671 with bevacizumab, 1,095 with ramucirumab, and 372 with aflibercept beta) received second-line FOLFIRI plus antiangiogenic drug treatment and were defined as the FOLFIRI plus antiangiogenic drug subpopulation.

### First-line and early recurrence populations

**Treatment sequences.** Patients in the first-line population, who started systemic therapy presumably for advanced disease, most commonly received first-line FOLFOX (leucovorin, 5-FU, oxaliplatin) or CAPOX (capecitabine and oxaliplatin) with bevacizumab (presumed *RAS*-mutant) and FOLFOX with panitumumab (presumed *RAS*-wild type) (Fig 2, S1A Table). The percentage of patients who transitioned from first- to second-line therapy was 56.0% for those with presumed *RAS*-mutant CRC and 71.3% for those with presumed *RAS*-wild type CRC. The most common second-line treatments in the first line population were FOLFIRI-based (Fig 2) and, in general, transition rates decreased over subsequent lines of therapy (Table 1).

Patients in the early recurrence population most commonly received CAPOX, capecitabine monotherapy, or UFT as adjuvant therapy followed by second-line chemotherapy doublet regimens with bevacizumab or an anti-EGFR antibody (i.e., panitumumab) (S1B Table). The percentage of patients who transitioned from second- to third-line therapy was 58.1% for those with presumed *RAS*-mutant CRC and 83.0% for those with presumed *RAS*-wild type CRC, and generally, transition rates decreased over later lines of therapy (Table 1).

Treatment sequences were summarised in the major regimen groups (S1A and S1B Fig). The most common transitions in the first-line population were from first-line oxaliplatin-based regimens to second-line irinotecan-based regimens (S1A Fig). The most common transitions in the early recurrence population were from adjuvant oxaliplatin-based regimens to second-line irinotecan-based regimens and from fluoropyrimidine monotherapy to oxaliplatin-based regimens (S1B Fig).

**Patient characteristics associated with prescription of second-line antiangiogenic drugs.** To understand the characteristics of patients who received antiangiogenic drugs during second-line therapy, we combined the first-line and early recurrence populations and assessed the patient demographics and clinical characteristics of those who received second-line therapy with and without antiangiogenic drugs (Table 2). The mean age of patients at the

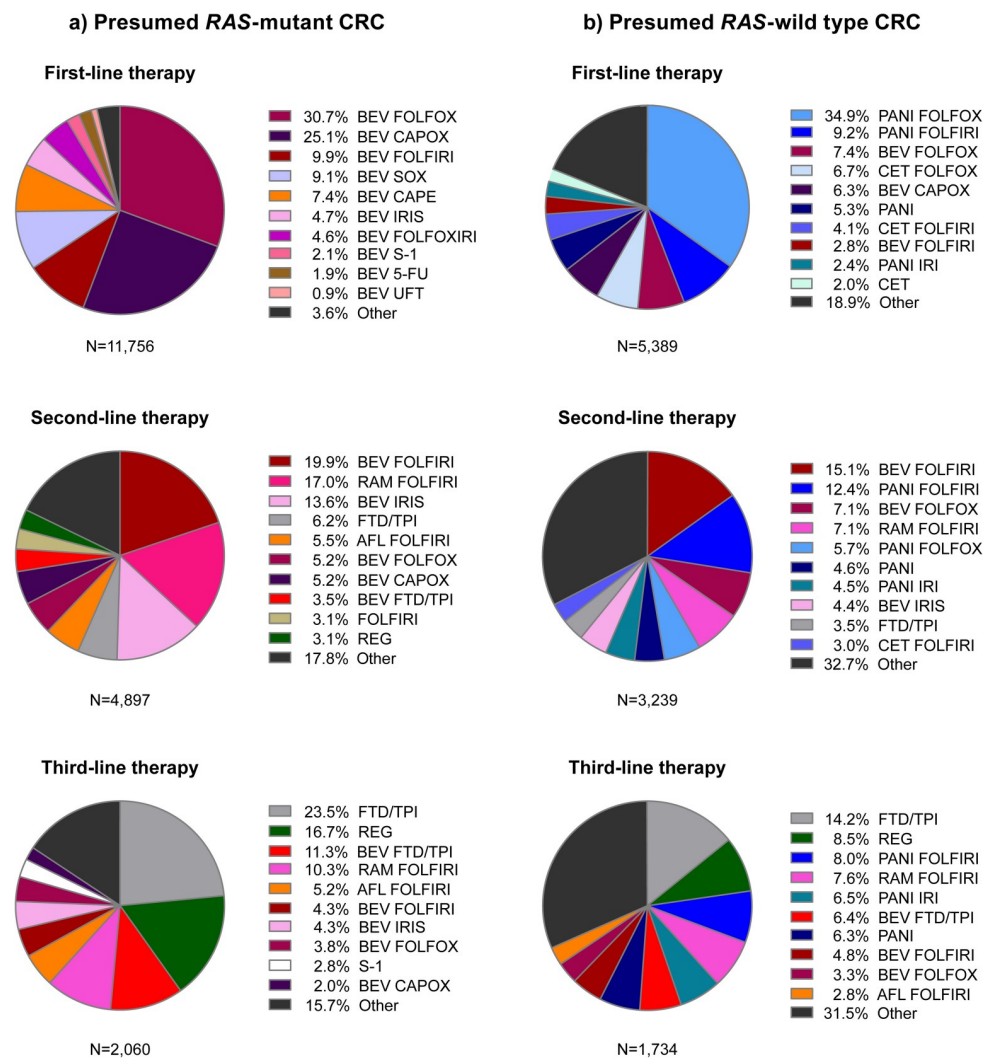

**Fig 2.** Treatment regimens in the first, second, and third lines for patients in the first-line population (started systemic therapy including biologics without prior evidence of early recurrence) who: a) had no anti-EGFR antibody prescription during the analysis period and were presumed to have *RAS*-mutant CRC, and b) had an anti-EGFR antibody prescription during the analysis period and were presumed to have *RAS*-wild type CRC. *RAS*, rat sarcoma viral oncogene homolog; CRC, colorectal cancer; BEV, bevacizumab; FOLFOX, leucovorin, 5-fluorouracil, and oxaliplatin; CAPOX, capecitabine and oxaliplatin; FOLFIRI, leucovorin, 5-fluorouracil, and irinotecan; SOX, S-1 and oxaliplatin; CAPE, capecitabine; IRIS, S-1 and irinotecan; FOLFOXIRI, leucovorin, 5-fluorouracil, oxaliplatin, and irinotecan; S-1, tegafur/gimestat/potassium otastat; 5-FU, 5-fluorouracil; UFT, uracil and tegafur; RAM, ramucirumab; FTD/TPI, trifluridine/tipiracil; AFL, aflibercept beta; REG, regorafenib; PANI, panitumumab; CET, cetuximab; IRI, irinotecan.

start of second-line therapy was approximately 66–67 years, more than half were men, and the mean BMI of patients was approximately 22 kg/m$^2$. Compared with patients without an anti-angiogenic drug prescription, fewer patients with any antiangiogenic drug prescription had presumed *RAS*-wild type CRC (approximately 26% vs 58%), and slightly more patients had right-sided CRC (approximately 34% vs 28%) and independent ADL (approximately 91% vs 86%).

In general, the characteristics and previous treatment of patients who received second-line antiangiogenic drugs (Table 2) were similar regardless of which combination chemotherapy was prescribed (i.e, with any chemotherapy or with FOLFIRI), except that fewer patients with

**Table 1. Transition rates between lines of therapy for the first-line and early recurrence populations.**

| | No anti-EGFR antibody prescription during the AP (presumed *RAS*-mutant CRC) | | | Anti-EGFR antibody prescription during the AP (presumed *RAS*-wild type CRC) | | |
|---|---|---|---|---|---|---|
| | Evaluable patients [a] | Transitioned patients | Rate, % | Evaluable patients [a] | Transitioned patients | Rate, % |
| **First-line population** | | | | | | |
| 1st to 2nd line | 8,739 | 4,897 | 56.0 | 4,542 | 3,239 | 71.3 |
| 2nd to 3rd line | 3,629 | 2,060 | 56.8 | 2,540 | 1,734 | 68.3 |
| 3rd to 4th line | 1,573 | 793 | 50.4 | 1,303 | 802 | 61.6 |
| 4th to 5th line | 558 | 239 | 42.8 | 594 | 333 | 56.1 |
| **Early recurrence population** | | | | | | |
| 2nd to 3rd line [b] | 991 | 576 | 58.1 | 283 | 235 | 83.0 |
| 3rd to 4th line | 406 | 241 | 59.4 | 186 | 130 | 69.9 |
| 4th to 5th line | 175 | 92 | 52.6 | 90 | 56 | 62.2 |

EGFR, epidermal growth factor receptor; AP, analysis period; *RAS*, rat sarcoma viral oncogene homolog; CRC, colorectal cancer.

[a] Patients with data available ≥60 days after the end of each line of therapy or patients who transitioned to the next line of therapy were included.

[b] Transition rate from adjuvant therapy to second-line therapy is 100% by definition for this population and is not included in this table.

second-line FOLFIRI received oral fluoropyrimidine in their previous line of therapy (25% vs 44%). Similar findings were observed when each of the antiangiogenic drugs were analysed separately (S2 Table). However, there were numerically fewer patients with presumed *RAS*-wild type CRC or anti-EGFR antibody use in the previous line of therapy, and numerically more patients with a bevacizumab, irinotecan, or FOLFOXIRI (leucovorin, 5-FU, oxaliplatin, irinotecan) prescription, in the ramucirumab and aflibercept beta groups than in the bevacizumab group.

### FOLFIRI plus antiangiogenic drug subpopulation

**Prescription details of second-line antiangiogenic drugs.** The mean and median duration of second-line FOLFIRI plus antiangiogenic drug treatment were 6.7 (standard error 0.15) months and 4.5 (95% CI 4.3, 4.8) months, respectively, and 66.2% of patients transitioned to third-line therapy (Table 3). Antiangiogenic drugs were prescribed for a median of 5 times during second-line treatment, 12.1% of patients had a dose reduction, and 53.9% of patients had at least one ≥21-day interval between prescriptions, which means a ≥7-day delay between prescriptions compared with the expected biweekly dosing schedule (Table 3). Although background adjustment could not be conducted to statistically compare individual drugs, the observed duration of second-line treatment and transition rates to third-line therapy were numerically higher in patients with a bevacizumab prescription compared with the other antiangiogenic drugs (S3A–S3C Table). Compared with the other antiangiogenic drugs, fewer patients who received bevacizumab had a dose reduction and fewer patients received bevacizumab only once during second-line therapy (S3A–S3C Table).

During second-line FOLFIRI plus antiangiogenic drug treatment, 11.7% of patients received only one dose of antiangiogenic drug. In the multivariate logistic regression analysis, the demographic factors significantly associated with patients receiving one dose of second-line antiangiogenic drug compared with ≥2 doses were older age (≥70 years, odds ratio 1.72 [95% CI 1.31, 2.26], p<0.0001) and not fully independent ADL (odds ratio 2.04 [95% CI 1.38, 3.03], p = 0.0004). Presumed *RAS*-wild type CRC (odds ratio 0.53 [95% CI 0.38, 0.75], p = 0.0003) and early recurrence (odds ratio 0.50 [95% CI 0.26, 0.95], p = 0.0333) were associated with ≥2 doses versus one dose.

**Table 2. Baseline demographics and clinical characteristics of patients with CRC in the combined first-line and early recurrence populations, categorised by use of antiangiogenic drugs as second-line therapy.**

| Variable | Second-line therapy | | |
|---|---|---|---|
| | Non-antiangiogenic drugs N = 3,215 | Any antiangiogenic drugs, with any chemotherapy N = 6,613 | Any antiangiogenic drugs, with FOLFIRI N = 3,138 |
| <200 beds | 159 (5.0) | 407 (6.2) | 202 (6.4) |
| 200–499 beds | 1,783 (55.5) | 3,768 (57.0) | 1,826 (58.2) |
| ≥500 beds | 1,273 (39.6) | 2,438 (36.9) | 1,110 (35.4) |
| Designated cancer hospital, n (%) | 2,494 (77.6) | 4,902 (74.1) | 2,253 (71.8) |
| Medical department where first 2nd-line therapy was prescribed, n (%) | | | |
| Internal medicine | 1,006 (31.3) | 1,802 (27.3) | 878 (28.0) |
| Surgery | 2,182 (67.9) | 4,737 (71.6) | 2,223 (70.8) |
| Others | 26 (0.8) | 68 (1.0) | 34 (1.1) |
| Unknown | 1 (0.03) | 6 (0.1) | 3 (0.1) |
| Age at start of 2nd line, mean (SD) | 67.2 (10.7) | 66.8 (10.4) | 66.0 (10.3) |
| Age ≥70 years at start of 2nd line, n (%) | 1,447 (45.0) | 2,918 (44.1) | 1,272 (40.5) |
| Sex: male, n (%) | 2,019 (62.8) | 3,877 (58.6) | 1,821 (58.0) |
| Presumed *RAS*-wild type CRC (received anti-EGFR antibody at any time), n (%) | 1,860 (57.9) | 1,709 (25.8) | 830 (26.5) |
| Left-sided colorectal cancer, n (%) | 2,389 (74.3) | 4,561 (69.0) | 2,162 (68.9) |
| Right-sided colorectal cancer, n (%) | 908 (28.2) | 2,246 (34.0) | 1,077 (34.3) |
| BMI, kg/m$^2$, mean (SD) [a] | 22.1 (3.8) | 22.3 (3.7) | 22.3 (3.7) |
| ADL–independent, n (%)[b] | 2,220 (86.0) | 4,939 (90.6) | 2,534 (92.1) |
| ADL–not independent. n (%) [b] | 363 (14.1) | 513 (9.4) | 217 (7.9) |
| Included in 1st-line or adjuvant therapy, n (%) | | | |
| Bevacizumab | 1,883 (58.6) | 4,370 (66.1) | 2,306 (73.5) |
| Anti-EGFR antibody | 745 (23.2) | 1,163 (17.6) | 585 (18.6) |
| Fluoropyrimidine, oral | 1,494 (46.5) | 2,897 (43.8) | 781 (24.9) |
| Fluoropyrimidine, i.v. | 1,481 (46.1) | 3,546 (53.6) | 2,465 (78.6) |
| Irinotecan | 730 (22.7) | 1,074 (16.2) | 342 (10.9) |
| FOLFOXIRI | 112 (3.5) | 190 (2.9) | 113 (3.6) |
| Treatment duration of 1st-line or adjuvant, days, median (IQR) | 159 (75–268) | 175 (98–283) | 176 (99–284) |

CRC, colorectal cancer; FOLFIRI, leucovorin, fluorouracil, and irinotecan; SD, standard deviation; *RAS*, rat sarcoma viral oncogene homolog; EGFR, endothelial growth factor receptor; BMI, body mass index; ADL, activities of daily living; i.v., intravenous; FOLFOXIRI, leucovorin, fluorouracil, oxaliplatin, and irinotecan; IQR, interquartile range.

[a] Non-antiangiogenic treatment, N = 2,659; any antiangiogenic treatment, N = 5,509; FOLFIRI + antiangiogenic drugs, N = 2,774.

[b] Non-antiangiogenic treatment, N = 2,583; any antiangiogenic treatment, N = 5,452; FOLFIRI + antiangiogenic drugs, N = 2,751.

Within the FOLFIRI plus antiangiogenic drug subpopulation, compared with patients who received ≥2 doses of antiangiogenic drugs, more patients who received one dose were prescribed antihypertensive medication (27.1% [n = 102] vs 20.4% [562], chi-square test p = 0.0035) or underwent quantitative proteinuria tests (12.7% [n = 48] vs 6.3% [n = 174], chi-square test p<0.0001) on the first day of their second-line antiangiogenic therapy. When each antiangiogenic drug was evaluated separately, these differences were more evident in patients who received ramucirumab or aflibercept beta than bevacizumab (data not shown). Similarly, numerically more patients who received ramucirumab or aflibercept beta were prescribed these concomitant drugs, anticholinergics, and proteinuria tests on the first day of their antiangiogenic therapy than those who received bevacizumab, irrespective of the number of antiangiogenic drug prescriptions (S4 Table).

**Table 3. Prescription characteristics and treatment continuation in the FOLFIRI plus antiangiogenic drug sub-population (second-line treatment).**

| Variable | Value |
|---|---|
| Duration of 2nd-line treatment with antiangiogenic drugs (months) [a] | N = 3,138 |
| Mean (SE) | 6.7 (0.15) |
| Median (95% CI) | 4.5 (4.3–4.8) |
| Prescription characteristics and transition rate in 2nd-line treatment with FOLFIRI plus antiangiogenic drugs | N = 2,430 [b] |
| Patients who transitioned to 3rd-line treatment, n (%) | 1,608 (66.2) |
| Number of antiangiogenic drug prescriptions, median (IQR) | 5 (3–9) |
| Patients with antiangiogenic drug dose reductions, n (%) | 295 (12.1) |
| Patients with antiangiogenic drug prescription gaps ≥21 days, n (%) | 1,310 (53.9) |
| Patients who used antiangiogenic drug once, n (%) | 283 (11.7) |

FOLFIRI, leucovorin, fluorouracil, and irinotecan; SE, standard error; CI, confidence interval; IQR, interquartile range.

[a] Duration was estimated using the Kaplan-Meier method. The mean survival time and its standard error were underestimated because the largest observation was censored and the estimation was restricted to the largest event time.

[b] Patients with data available ≥60 days after the end of second-line therapy or patients who transitioned to third-line therapy were included in this analysis.

**Factors associated with overall treatment duration from second-line FOLFIRI plus anti-angiogenic drugs.** Multivariate Cox regression analysis was undertaken to investigate the factors associated with the overall duration of all antitumor drug therapy from the start of second-line treatment with FOLFIRI plus antiangiogenic drugs (Table 4). The factors significantly associated with longer treatment duration were left-sided CRC, presumed *RAS*-wild type CRC, treatment with oral fluoropyrimidine in the previous line of therapy, early recurrence, the presence of concomitant qualitative tests for proteinuria, and use of antihypertensives during second-line therapy. In contrast, the factors significantly associated with shorter treatment duration were designated cancer hospital, low BMI and not fully independent ADL. In general, similar trends were observed in the subgroup analyses of patients with left-sided CRC (S5A Table), right-sided CRC (S5B Table), presumed *RAS*-wild type CRC (S5C Table), and presumed *RAS*-mutant CRC (S5D Table). In addition, findings from the subgroup analyses of each antiangiogenic drug in combination with FOLFIRI (S6A–S6C Table) were similar to those of the entire FOLFIRI plus antiangiogenic drug subpopulation (Table 4).

## Discussion

### Treatment sequences

Findings from this observational cohort study showed that treatment of patients with advanced CRC in Japan is consistent with both international [18, 19] and Japanese treatment guidelines [4]. The most common first-line therapies for advanced disease were FOLFOX or CAPOX with bevacizumab (presumed *RAS*-mutant) and FOLFOX with panitumumab (presumed *RAS*-wild type). The most common second-line treatments were FOLFIRI-based regimens.

An important finding from this study was that transition rates to subsequent lines of therapy were lower than those observed in randomised controlled trials conducted in Japan [20, 21], which suggests that a significant proportion of patients in Japan do not undergo post-

**Table 4. Multivariate Cox regression analysis for the factors associated with overall treatment continuation from the start of second-line therapy to the end of all antitumor drug therapies in the FOLFIRI plus antiangiogenic drug subpopulation.**

| Covariate | Hazard ratio | 95% CI | p-value |
|---|---|---|---|
| Designated cancer hospital (yes vs no) | 1.13 | 1.01–1.26 | 0.0336 |
| $\geq$70 vs <70 years at start of $2^{nd}$-line therapy | 1.06 | 0.96–1.18 | 0.2431 |
| Sex: male vs female | 1.05 | 0.95–1.16 | 0.3712 |
| Left-sided colorectal cancer (yes vs no) | 0.85 | 0.76–0.95 | 0.0035 |
| Presumed *RAS*-wild type (yes vs no) | 0.64 | 0.57–0.72 | <0.0001 |
| BMI $\leq$18.5 kg/m$^2$ vs >18.5 kg/m$^2$ | 1.31 | 1.14–1.51 | 0.0001 |
| ADL (not independent vs independent) | 1.36 | 1.14–1.61 | 0.0006 |
| Oral fluoropyrimidine in previous line of therapy (yes vs no) | 0.80 | 0.71–0.91 | 0.0005 |
| Irinotecan in previous line (yes vs no) | 1.16 | 0.98–1.38 | 0.0929 |
| Duration of previous line of therapy $\geq$180 days vs <180 days | 0.91 | 0.82–1.01 | 0.0761 |
| Early recurrence (yes vs no) | 0.68 | 0.56–0.82 | <0.0001 |
| Concomitant procedures and medications during $2^{nd}$-line therapy (yes vs no) | | | |
| Qualitative proteinuria tests | 0.73 | 0.65–0.81 | <0.0001 |
| Quantitative proteinuria tests | 0.90 | 0.79–1.04 | 0.1458 |
| Antihypertensives | 0.89 | 0.80–0.99 | 0.0251 |
| Anticholinergics | 0.91 | 0.81–1.03 | 0.1502 |
| Anticoagulants | 0.94 | 0.75–1.18 | 0.5917 |

FOLFIRI, leucovorin, fluorouracil, and irinotecan; CI, confidence interval; *RAS*, rat sarcoma viral oncogene homolog; BMI, body mass index; ADL, activities of daily living; EGFR, endothelial growth factor receptor.

2,735 patients who started FOLFIRI plus antiangiogenic drug as second-line and had ADL and BMI data available from baseline period before second-line were included in this analysis.

progression therapy in real-world practice. This finding is not unexpected considering the highly selected population in clinical trials compared with real-world practice, but implies important unmet needs, as several studies in patients with CRC [22] and other cancer types [23, 24] have demonstrated improved or prolonged survival in patients who undergo post-progression therapy. However, the MDV database does not include information about treatment line completion. Therefore, transition rates, which were estimated using an analysis algorithm and were calculated only for those who had completed their previous line of therapy, may not be fully accurate.

## Treatment choices for second-line therapy

A key challenge for clinical decision-making is which treatment regimen to select for patients with advanced disease who cannot tolerate first-line therapy or when first-line therapy has failed [25]. In this study, second-line therapy that included antiangiogenic drugs, regardless of the chemotherapy backbone (FOLFIRI vs any chemotherapy), was prescribed more frequently in patients with presumed *RAS*-mutant CRC, and had right-sided CRC and independent ADL. Compared with patients who received any chemotherapy regimen plus antiangiogenic drugs, fewer patients who received FOLFIRI plus antiangiogenic drugs had used oral fluoropyrimidine in their previous line of therapy, possibly because these patients found it difficult, either medically or psychologically, to switch to intravenous fluoropyrimidine included in FOLFIRI, when proceeding to second-line therapy.

There was a difference in prior therapy choice among the antiangiogenic drug subgroups within the FOLFIRI plus antiangiogenic drug subpopulation. Compared to patients with a second-line bevacizumab, more patients with second-line ramucirumab or aflibercept beta had

received bevacizumab in their previous line of therapy. As clinicians are likely to have more experience with bevacizumab, it is possible that if bevacizumab is not used as first-line therapy, it is selected as the first antiangiogenic drug for second-line therapy. For ramucirumab, the eligibility criteria of the RAISE trial, which only enrolled patients who received first-line bevacizumab (without an anti-EGFR antibody) [5], may have influenced clinical decision-making. In contrast, the VELOUR trial for aflibercept beta was not restricted to patients who had used bevacizumab previously [7] and, consistent with our results, most patients (83.3%) in the Japanese trial for aflibercept beta had received first-line bevacizumab [6].

## Prescription details of treatment with second-line antiangiogenic drugs

Overall, the duration of treatment with second-line FOLFIRI plus antiangiogenic drugs and the rate of transition to subsequent lines of therapy were comparable with the results reported from randomised clinical trials for antiangiogenic drugs [5, 7, 26] and from real-world clinical practice [27]. However, treatment duration and transition rates are slightly higher for patients enrolled in the randomised trials compared with real-world studies, which is to be expected because of the eligibility criteria and close monitoring of patients in trials.

In this study, as many as 11.7% of patients who were treated with second-line FOLFIRI plus antiangiogenic drugs received only one dose before terminating antiangiogenic drug treatment. The factors associated with patients receiving only one dose compared with multiple doses were older age (≥70 years) and not fully independent ADL, which suggests that the smaller number of doses of second-line antiangiogenic drugs in this study (median 4–6 doses) compared with randomised trials (median 8 doses [5, 6, 28]) is partly because of the older population. Generally, management of older patients with CRC is challenging, and there is limited information on the ability of older patients to tolerate targeted therapies [29]. In this real-world study, patients in each of the antiangiogenic subgroups were consistently older (by 4–6 years) than those in randomised trials [5, 7, 30]. Conversely, we found that presumed *RAS*-wild type CRC and early recurrence were associated with ≥2 doses of antiangiogenic drugs. The proportion of patients with presumed *RAS*-wild type CRC in this study (27%) was much lower than in the phase III trials for each antiangiogenic drug (51–58% *KRAS*-wild type [5, 31, 32]). Considering that *RAS* mutation status is a known prognostic factor for advanced CRC [33], it is reasonable to speculate that the lower proportion of presumed *RAS*-wild type patients in this real-world study might also have contributed to the smaller number of antiangiogenic drug prescriptions compared with randomised trials. However, because our definition of *RAS* status was presumed based on the prescription for anti-EGFR antibody, there was a potential for misclassification of *RAS* status.

We also observed that more patients who received only one dose of FOLFIRI plus antiangiogenic drugs also received concomitant quantitative proteinuria tests and prescriptions for antihypertensives at the start of their antiangiogenic drug therapy. Proteinuria and hypertension are known adverse effects associated with antiangiogenic drugs [34–36] and, as the proteinuria tests and antihypertensives were prescribed at the same time as the first antiangiogenic drug prescription, it is possible that they were associated with pre-existing complications related to patients' previous therapies. However, we could not confirm such causal relationships because of the limitations of retrospective database studies [37].

Several differences in patient characteristics, treatment history, and the need for dose reductions were observed between each of the antiangiogenic drugs. Compared with second-line FOLFIRI plus bevacizumab, fewer patients with the other antiangiogenic drugs were presumed to have *RAS*-wild type CRC, more had used bevacizumab, irinotecan, and FOLFOXIRI in their previous line of therapy, and more were prescribed anticholinergics, antihypertensives

and proteinuria tests at the start of second-line treatment. In addition, fewer patients who received FOLFIRI plus bevacizumab experienced dose reductions and were less likely to terminate their prescription after only one dose. While the potential reasons for these differences are not known, it is likely that the differences in patients' backgrounds, the safety profiles of each antiangiogenic drug, and/or clinician experience with each drug were contributing factors. Together with the observation that treatment duration differed between the second-line antiangiogenic drugs, these findings suggest that further investigation into the associations between patient background, treatment choice, and treatment outcome is warranted.

## Factors associated with overall treatment duration from second-line FOLFIRI plus antiangiogenic drugs

In this study, we explored the treatments and clinical factors associated with overall duration of sequential therapy, starting from treatment with second-line FOLFIRI plus antiangiogenic drugs. Left-sided and presumed *RAS*-wild type CRC in Japanese patients in our study were associated with longer treatment duration, consistent with other reports that these are positive prognostic factors [38]. As expected, patients with poor health status, as indicated by low BMI and not independent ADL, had shorter treatment durations. The previous treatment factor associated with patients continuing sequential therapy was treatment with oral fluoropyrimidines in their previous lines of therapy.

We also showed that the presence of qualitative tests for proteinuria and use of antihypertensives during second-line treatment were associated with longer treatment continuation, starting from second-line FOLFIRI plus antiangiogenic drugs. As quantitative proteinuria tests are often conducted as part of daily monitoring of patients treated with these drugs, this result might imply that adverse event monitoring for proteinuria during treatment is important with antiangiogenic drugs. Although hypertension has been reported to be associated with the effectiveness of bevacizumab in patients with CRC [39] and ramucirumab in patients with gastric cancer [40, 41], this association is not unanimously accepted because it has not been reported in patients receiving ramucirumab treatment for CRC [42]. Because of the nature of retrospective studies, it is not possible to determine whether a causal relationship exists between longer overall treatment duration and proteinuria tests or antihypertensive prescriptions. In addition, it is not clear whether patients actually experienced adverse events when these procedures for adverse event management were conducted.

## Strengths and limitations

The main strength of this study is the use of an administrative claims database covering a large number of Japanese DPC hospitals, which maximised the number of patients with CRC being treated in clinical practice that could be included in the study. Because of the nature of administrative databases, the MDV database is expected to include hospitals that do not actively publish clinical studies in the peer-reviewed literature. As such, use of the MDV database can provide greater insight into real-world treatment practices in Japan that are not limited to reports from active researchers or to findings from large prospective clinical trials. This is particularly relevant to patient populations who are underrepresented or not represented in prospective clinical trials, such as older patients, and, in the case of ramucirumab, those with prior first-line anti-EGFR antibody therapy. In addition, we were able to assess the association of patient care/monitoring procedures such as proteinuria tests and concomitant therapies that are conducted at the discretion of treating hospitals and where their use is not as strictly defined compared with clinical trials.

Despite the value of conducting a study in a real-world population, there are several limitations associated with the use of the database that may affect the interpretation of the findings. First, the MDV database is not population based, and patients from smaller or specialist medical centres were not included. In addition, by limiting the study population to those who received bevacizumab, cetuximab, or panitumumab (first-line population), patients not eligible to use biological therapies but who received systemic therapies for advanced diseases were excluded from the study. Therefore, the findings may not be applicable to the total population with CRC in Japan. Second, because patients cannot be tracked between hospitals, those who received treatment at more than one hospital in the MDV database may have been counted twice. Third, the MDV database largely lacks clinical information about patients and diseases, including treatment outcomes. For example, *RAS* mutation status is not available in the database, so patients with an anti-EGFR antibody during the analysis period were presumed to have *RAS*-wild type CRC, and the potential for patients to change *RAS* mutation status during their treatment sequence was not considered. This lack of clinical information significantly hampers attempts to adjust the analyses for patient demographics and clinical characteristics. Finally, the MDV database does not include information about individual treatment regimens or lines of therapy. Therefore, several assumptions needed to be made to define the treatment sequences, which may not be entirely accurate. In addition, the presence of a prescription claim in the MDV database may not be representative of actual treatment dose, especially for oral drugs.

## Conclusions

This is the first real-world investigation of treatment sequences for patients with advanced CRC in Japan using a large administrative database. The results of this study suggest that treatment of advanced CRC in Japan is consistent with global and local treatment guidelines. However, a considerable proportion of patients do not transition to subsequent lines of therapy, despite the increasing number of new treatment options. Our findings also suggest that in addition to antitumor drug sequence, better general health status (specifically related to BMI and ADL), management of hypertension, and testing for proteinuria during treatment are associated with continuation of sequential therapy that includes antiangiogenic drugs. As no causal relationship can be concluded from this retrospective study, further studies to evaluate causal relationships are warranted.

## Supporting information

**S1A Fig. Treatment sequences for the first-line population.**
(PDF)

**S1B Fig. Treatment sequences for the early recurrence population.**
(PDF)

**S1A Table. Five most common treatment regimens in the first, second, and third lines for patients who started systemic therapy including biologics without prior evidence of early recurrence (first-line population).**
(PDF)

**S1B Table. Five most common treatment regimens in the adjuvant, second, and third lines for patients who underwent colorectal surgery and experienced early recurrence after adjuvant therapy (early recurrence population).**
(PDF)

**S2 Table. Baseline demographics and clinical characteristics of patients with CRC in the combined first-line and early recurrence populations who received FOLFIRI in combination with antiangiogenic drugs.**
(PDF)

**S3A Table. Prescription characteristics and treatment continuation in the FOLFIRI plus bevacizumab population (second-line treatment).**
(DOCX)

**S3B Table. Prescription characteristics and treatment continuation in the FOLFIRI plus ramucirumab population (second-line treatment).**
(DOCX)

**S3C Table. Prescription characteristics and treatment continuation in the FOLFIRI plus aflibercept beta population (second-line treatment).**
(DOCX)

**S4 Table. Concomitant procedures and medications received on the first date of a second-line antiangiogenic therapy in the FOLFIRI plus antiangiogenic drug subpopulation.**
(PDF)

**S5A Table. Multivariate Cox regression analysis for the factors associated with overall treatment continuation from the start of second-line therapy to the end of all antitumor drug therapies in the FOLFIRI plus antiangiogenic drug subpopulation, for patients with left-sided CRC.**
(PDF)

**S5B Table Multivariate Cox regression analysis for the factors associated with overall treatment continuation from the start of second-line therapy to the end of all antitumor drug therapies in the FOLFIRI plus antiangiogenic drug subpopulation, for patients with right-sided CRC.**
(PDF)

**S5C Table. Multivariate Cox regression analysis for the factors associated with overall treatment continuation from the start of second-line therapy to the end of all antitumor drug therapies in the FOLFIRI plus antiangiogenic drug subpopulation, for patients with presumed *RAS*-wild type CRC.**
(PDF)

**S5D Table. Multivariate Cox regression analysis for the factors associated with overall treatment continuation from the start of second-line therapy to the end of all antitumor drug therapies in the FOLFIRI plus antiangiogenic drug subpopulation, for patients with presumed *RAS*-mutant CRC.**
(PDF)

**S6A Table. Multivariate Cox regression analysis for the factors associated with overall treatment continuation from the start of second-line therapy to the end of all antitumor drug therapies in the FOLFIRI plus bevacizumab population.**
(PDF)

**S6B Table. Multivariate Cox regression analysis for the factors associated with overall treatment continuation from the start of second-line therapy to the end of all antitumor drug therapies in the FOLFIRI plus ramucirumab population.**
(PDF)

**S6C Table. Multivariate Cox regression analysis for the factors associated with overall treatment continuation from the start of second-line therapy to the end of all antitumor drug therapies in the FOLFIRI plus aflibercept beta population.**
(PDF)

## Acknowledgments

Medical writing assistance was provided by Serina Stretton, PhD, CMPP, and Rebecca Lew, PhD, CMPP, of ProScribe–Envision Pharma Group. ProScribe's services complied with international guidelines for Good Publication Practice (GPP3).

## Author Contributions

**Conceptualization:** Eiji Shinozaki, Akitaka Makiyama, Yoshinori Kagawa, Hironaga Satake, Yoshinori Tanizawa, Zhihong Cai, Yongzhe Piao.

**Formal analysis:** Yoshinori Tanizawa, Zhihong Cai.

**Funding acquisition:** Yoshinori Tanizawa, Yongzhe Piao.

**Investigation:** Eiji Shinozaki, Akitaka Makiyama, Yoshinori Kagawa, Hironaga Satake, Yoshinori Tanizawa, Zhihong Cai, Yongzhe Piao.

**Methodology:** Eiji Shinozaki, Akitaka Makiyama, Yoshinori Kagawa, Hironaga Satake, Yoshinori Tanizawa, Zhihong Cai, Yongzhe Piao.

**Project administration:** Yoshinori Tanizawa.

**Supervision:** Yongzhe Piao.

**Validation:** Zhihong Cai.

**Visualization:** Eiji Shinozaki, Akitaka Makiyama, Yoshinori Kagawa, Hironaga Satake, Yoshinori Tanizawa, Zhihong Cai, Yongzhe Piao.

**Writing – original draft:** Eiji Shinozaki, Akitaka Makiyama, Yoshinori Kagawa, Hironaga Satake, Yoshinori Tanizawa, Zhihong Cai, Yongzhe Piao.

**Writing – review & editing:** Eiji Shinozaki, Akitaka Makiyama, Yoshinori Kagawa, Hironaga Satake, Yoshinori Tanizawa, Zhihong Cai, Yongzhe Piao.

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
