## [Decision Letter · Decision Letter 0]

1 Sep 2020

PONE-D-20-22549

Treatment sequences of patients with colorectal cancer treated with ramucirumab in Japan: A retrospective observational study using an administrative database

PLOS ONE

Dear Dr. Tanizawa,

Thank you for submitting your manuscript to PLOS ONE. After careful consideration, we feel that it has merit but does not fully meet PLOS ONE’s publication criteria as it currently stands. Therefore, we invite you to submit a revised version of the manuscript that addresses the points raised during the review process.

We look forward to receiving your revised manuscript.

Kind regards,

Takeshi Nagasaka, M.D., Ph. D.

Academic Editor

PLOS ONE

Additional Editor Comments:

As the revisers’ comments, this study is too much focus on Ramucirumab, and did not evaluate aflibercept or bevacizumab with FOLFIRI as second line chemotherapy. Moreover, Yoshinori

Tanizawa, Zhihong Cai, and Yongzhe Piao are hired Eli Lilly company which sell RAMUCIRUMAB. The authors should be re-evaluating all of issues in this study for these regimens, and please discuss the results, objectively.

The MDV database has many biases, as the reviewer 2 mentioned. The authors make too emphasis about RAMUCIRUMAB.

However, If the authors re-edit this data not only for RAMUCIRUMAB but also other two anti-VEGF antibodies, this article will be worthful for giving information to clinicians in relation to considering 2nd line regimens.

Therefore, I decide this article as Major revision.

Journal Requirements:

2. In your Methods section, please provide additional information about the demographic details of your participants. Please ensure you have provided sufficient details to replicate the analyses such as a table of relevant demographic details.

"I have read the journal's policy and the authors of this manuscript have the following competing interests:

ES has received honoraria from Taiho Pharmaceutical Co., Ltd., Takeda Pharmaceutical Co., Ltd., Chugai Pharmaceutical Co., Ltd., Eli Lilly Japan Co., Ltd., Merck Bio Pharma Co., Ltd., Sanofi Co., Ltd., Taiho Pharmaceutical Co., Ltd., and Yakult Honsha Co., Ltd.

YK has received honoraria from Bayer Co., Ltd., Chugai Pharmaceutical Co., Ltd., Yakult Honsha Co., Ltd., Sanofi Co., Ltd., Eli Lilly Japan Co., Ltd., Taiho Pharmaceutical Co., Ltd., Takeda Pharmaceutical Co., Ltd., and Merck Co., Ltd.

AM has received honoraria from Eli Lilly Japan Co., Ltd., Chugai Pharmaceutical Co., Ltd., and Takeda Pharmaceutical Co., Ltd.

HS has received research funding from Ono Pharmaceutical Co., Ltd., Taiho Pharmaceutical Co., Ltd., and Takeda Pharmaceutical Co., Ltd., and honoraria from Bayer Co., Ltd., Bristol-Myers Squibb Co., Ltd., Chugai Pharmaceutical Co., Ltd., Daiichi Sankyo Co., Ltd., Eli Lilly Japan Co., Ltd., Merck Bio Pharma Co., Ltd., MSD Co., Ltd., Ono Pharmaceutical Co., Ltd., Sanofi Co., Ltd., Taiho Pharmaceutical Co., Ltd., Takeda Co., Ltd., and Yakult Honsha Co., Ltd.

YT, ZC, and YP are employees and/or minor shareholders of Eli Lilly Japan K.K."

We note that one or more of the authors have an affiliation to the commercial funders of this research study : Eli Lilly Japan K.K.

3.1. Please provide an amended Funding Statement declaring this commercial affiliation, as well as a statement regarding the Role of Funders in your study. If the funding organization did not play a role in the study design, data collection and analysis, decision to publish, or preparation of the manuscript and only provided financial support in the form of authors' salaries and/or research materials, please review your statements relating to the author contributions, and ensure you have specifically and accurately indicated the role(s) that these authors had in your study. You can update author roles in the Author Contributions section of the online submission form.

3.2. Please also provide an updated Competing Interests Statement declaring this commercial affiliation along with any other relevant declarations relating to employment, consultancy, patents, products in development, or marketed products, etc.  

Reviewers' comments:

Reviewer's Responses to Questions

**Comments to the Author**

1. Is the manuscript technically sound, and do the data support the conclusions?

Reviewer #1: Partly

Reviewer #2: No

Reviewer #3: Yes

2. Has the statistical analysis been performed appropriately and rigorously? 

Reviewer #1: No

Reviewer #2: Yes

Reviewer #3: Yes

3. Have the authors made all data underlying the findings in their manuscript fully available?

Reviewer #1: Yes

Reviewer #2: Yes

Reviewer #3: Yes

4. Is the manuscript presented in an intelligible fashion and written in standard English?

Reviewer #1: Yes

Reviewer #2: Yes

Reviewer #3: Yes

5. Review Comments to the Author

Reviewer #1: The authors described the treatment sequences for advanced colorectal cancer (CRC) and the use of ramucirumab in Japan, and to explore factors associated with overall treatment continuation with antitumor drugs.

Factors significantly associated with shorter treatment continuation from second-line ramucirumab were low body mass index, not fully independent ADL, and previous irinotecan treatment. Factors significantly associated with longer treatment continuation were left-sided CRC, previous use of oral fluoropyrimidines or anti-EGFR antibody, longer duration of the previous therapy line (≥180 days), and use of qualitative tests for proteinuria and antihypertensives during second-line therapy.

The authors concluded that management of hypertension and testing for proteinuria are important for successful sequential therapy that includes antiangiogenic drugs.

I think that the meaningful table is only Table5. However, background is so mixed, and only univariate data were shown. I think that these analyses should be separated based on RAS status and sidedness at first, and multivariate analysis should be tried.

In conclusion, based on Table5, the authors mentioned that "management of hypertension and testing for proteinuria are important for successful sequential therapy that includes antiangiogenic drugs."

However, there are no information about the real information about hypertension and proteinuria in the database. I think that it is impossible to produce this conclusion. I think that the checkup of proteinuria and blood pressure is essential for the treatment of CRC using ramucirumab. The doctor who do not check these parameters may not be able to manage any complications of chemotherapy, leading to bias.

Reviewer #2: The manuscript is well written and shows a lot of real-world data of treatment sequence for CRC patients in Japan using a large administrative database. However, as authors mentioned in limitations session, the using database is not population based and may be far from general clinical data. For example, I recognized that the transition rate from 1st to 2nd line for CRC patients was almost 80%. On the other hand, this study showed the transition rate from 1st to 2nd line for presumed RAS-murtant CRC was 56.0% in Table 1. I think this MDV database lacks detailed clinical information.

Another point to mention, the authors suggest that management of hypertension and testing for proteinuria during treatment are important for successful continuation of sequential therapy that includes antiangiogenic drugs in conclusion session. Generally, patients who continue treatment for a long time have an increased risk of developing adverse events. So, it is difficult to say that management of hypertension and testing for proteinuria are important for successful continuation of treatment as conclusion. This conclusion could mislead the reader.

Reviewer #3: In this study, the authors, Shinozaki E et al, assessed the treatment sequences and use of ramucirumab in real-world data using MDV database to clarify the treatment sequences of patients with advanced CRC in Japan and to explore the clinical and treatment-related factors associated with treatment continuation with ramucirumab. Although this study includes several novel and interesting points, totally this paper is too much complicated and very difficult to read and understand. Several points need to be addressed for publication of “Plos One”.

My major concern is described below

1. Currently, one of the most relevant issues in CRC chemotherapy is the decision making of anti-VEGF drugs in second line treatment with FOLFIRI. However, this study is too much focus on Ramucirumab, and did not evaluate aflibercept or bevacizumab with FOLFIRI as second line chemotherapy. The authors should need to be re-evaluate all of issues in this study for these regimens, and please discuss the results. It is really helpful and clinically meaningful for clinicians.

2. This manuscript is too much complicated, and it is very difficult to read and understand due to assessment of several aspects using complicated and not perfect MDV database. Therefore, it is better to re-write to focus more clinically relevant points to easy to understand for readers.

6. PLOS authors have the option to publish the peer review history of their article (what does this mean?). If published, this will include your full peer review and any attached files.

Reviewer #1: No

Reviewer #2: No

Reviewer #3: No

---

## [Author Response · Author response to Decision Letter 0]

5 Nov 2020

Yoshinori Tanizawa

Medicines Development Unit-Japan,

Eli Lilly Japan K.K., LILLY PLAZA ONE

5-1-28 Isogamidori, Chuo-Ku

Kobe, Hyogo 651-0086, Japan

Tel: +81-80-9305-2141

E-mail: tanizawa_yoshinori@lilly.com

30 October 2020

Dr Joerg Heber

Editor-in-Chief, PLOS ONE

jheber@plos.org

Dear Dr Heber,

Re: Resubmission of Manuscript ID PONE-D-20-22549

On behalf of my coauthors, I am pleased to submit to PLOS ONE our revised manuscript, which is newly entitled “Treatment sequences of patients with advanced colorectal cancer and use of second-line FOLFIRI with antiangiogenic drugs in Japan: A retrospective observational study using an administrative database”

Thank you for your positive and constructive feedback on our original submission (your email dated 02 September 2020). We have responded to each of the reviewers’ and editor’s comments (please see below responses to each comment). 

We thank you for your willingness to consider a revised version of our manuscript and look forward to your earliest response.

Sincerely,

Yoshinori Tanizawa

Eli Lilly Japan K.K.

RESPONSE TO COMMENTS FROM JOURNAL

1. Please ensure that your manuscript meets PLOS ONE's style requirements, including those for file naming

We confirm that the manuscript meets PLOS ONE’s style requirements.

2. In your Methods section, please provide additional information about the demographic details of your participants. Please ensure you have provided sufficient details to replicate the analyses such as a table of relevant demographic details.

This study was designed to investigate treatment sequences of patients with advanced CRC in Japan using an administrative database. Therefore, the patient data included in this database provide a snapshot for a specific point in time, and cannot be replicated. The eligibility criteria describing which patients were included are reported in detail in the methods section (Study Populations) and in Figure 1. The baseline demographics and clinical characteristics of each of the datasets analysed are reported in Table 2 and S2 Table.

3.1. Please provide an amended Funding Statement declaring this commercial affiliation, as well as a statement regarding the Role of Funders in your study.

The amended Funding Statement, including the Role of Funders is below.

This study was sponsored by Eli Lilly Japan K.K., manufacturer/licensee of ramucirumab. The funder provided support in the form of salaries for authors (YT, ZC, and YP) and was involved in the study design, data collection, data analysis, preparation of the manuscript, and in the decision to publish.

3.2. Please also provide an updated Competing Interests Statement declaring this commercial affiliation along with any other relevant declarations relating to employment, consultancy, patents, products in development, or marketed products, etc. 

An updated Competing Interests Statement is below.

ES has received honoraria from Taiho Pharmaceutical Co., Ltd., Takeda Pharmaceutical Co., Ltd., Chugai Pharmaceutical Co., Ltd., Eli Lilly Japan Co., Ltd., Merck Bio Pharma Co., Ltd., Sanofi Co., Ltd., Taiho Pharmaceutical Co., Ltd., and Yakult Honsha Co., Ltd.

YK has received honoraria from Bayer Co., Ltd., Chugai Pharmaceutical Co., Ltd., Yakult Honsha Co., Ltd., Sanofi Co., Ltd., Eli Lilly Japan Co., Ltd., Taiho Pharmaceutical Co., Ltd., Takeda Pharmaceutical Co., Ltd., and Merck Co., Ltd. 

AM has received honoraria from Eli Lilly Japan Co., Ltd., Chugai Pharmaceutical Co., Ltd., and Takeda Pharmaceutical Co., Ltd.

HS has received research funding from Ono Pharmaceutical Co., Ltd., Taiho Pharmaceutical Co., Ltd., and Takeda Pharmaceutical Co., Ltd., and honoraria from Bayer Co., Ltd., Bristol-Myers Squibb Co., Ltd., Chugai Pharmaceutical Co., Ltd., Daiichi Sankyo Co., Ltd., Eli Lilly Japan Co., Ltd., Merck Bio Pharma Co., Ltd., MSD Co., Ltd., Ono Pharmaceutical Co., Ltd., Sanofi Co., Ltd., Taiho Pharmaceutical Co., Ltd., Takeda Co., Ltd., and Yakult Honsha Co., Ltd.

This study was sponsored by Eli Lilly Japan K.K., manufacturer/licensee of ramucirumab. YT, ZC, and YP are employees and/or minor shareholders of Eli Lilly Japan K.K. This does not alter our adherence to PLOS ONE policies on sharing data and materials, but the data used for this study are owned by a third party (Medical Data Vision Co. Ltd.).

4 a) If there are ethical or legal restrictions on sharing a de-identified data set, please explain them in detail (e.g., data contain potentially identifying or sensitive patient information) and who has imposed them (e.g., an ethics committee). Please also provide contact information for a data access committee, ethics committee, or other institutional body to which data requests may be sent.

The updated data sharing statement is below.

The data in this study belong to Medical Data Vision Co., Ltd (http://www.mdv.co.jp/) and were used under licence, funded by Eli Lilly Japan K.K. As the data are not publicly available, researchers looking to access the data used in this study should contact Medical Data Vision Co., Ltd via their website (http://www.mdv.co.jp/ [Japanese] or https://en.mdv.co.jp/ [English]).

RESPONSE TO COMMENTS FROM PEER REVIEW

Editor

Comment 1: 

This study is too much focus on Ramucirumab, and did not evaluate aflibercept or bevacizumab with FOLFIRI as second line chemotherapy. Moreover, Yoshinori Tanizawa, Zhihong Cai, and Yongzhe Piao are hired Eli Lilly company which sell RAMUCIRUMAB. The authors should be re-evaluating all of issues in this study for these regimens, and please discuss the results, objectively. 

The MDV database has many biases, as the reviewer 2 mentioned. The authors make too emphasis about RAMUCIRUMAB.

However, If the authors re-edit this data not only for RAMUCIRUMAB but also other two anti-VEGF antibodies, this article will be worthful for giving information to clinicians in relation to considering 2nd line regimens.

Authors’ Response:

We thank the editor for this comment. As noted in the responses to Reviewer 3 below, we have revised the main analysis population from FOLFIRI plus ramucirumab to FOLFIRI plus any antiangiogenic drug available in Japan (ie, bevacizumab, ramucirumab, and aflibercept beta), to avoid focusing on a particular drug. The title of the manuscript has been revised to reflect this change and the main results now report findings for FOLFIRI plus any antiangiogenic drug. These modifications did not substantially change our results or interpretation of the findings. Also, following your thoughtful recommendation to re-evaluate all of the issues in the study for all of the three treatment regimens, we have now included subgroup analyses for each of the antiangiogenic drugs as supplemental information (Supplementary Tables S2, S3a-c, S4, S6a-c), and objectively discussed similarities and differences among the three regimens in the Results and the Discussion. Table S4, which is new, summarizes the potential differences among patients who initiated each of the three regimens and provides information that may help inform clinicians’ decisions on which second-line treatment regimen to prescribe for their patients.

With regard to the biases in the database, especially the lack of clinical information that Reviewer 2 pointed out, although the limitations of the MDV database had been described in the Discussion previously, we have added a new paragraph to the Introduction that addresses the limitations of administrative database research generally, and we have expanded the relevant sections in the Discussion to ensure that the limitations to specific points are clearly highlighted. Please see responses to Reviewer 2 for more details.

With regard to the potential conflicts of interest of 3 of the authors, these have been fully disclosed in the manuscript and the focus on ramucirumab in the study has now been removed.

Reviewer 1

Comment 1:

I think that the meaningful table is only Table 5. However, background is so mixed, and only univariate data were shown. I think that these analyses should be separated based on RAS status and sidedness at first, and multivariate analysis should be tried.

Authors’ Response:

Thank you for this suggestion. We have added the requested subgroups (RAS status and CRC sidedness) to the analysis (see supplementary Tables S5a-d) and these data have been described in the Results section on page 18. 

With regard to the previous Table 5, which is now Table 4, the regression analysis was multivariate. We have modified the Abstract (page 3), Methods (page 11), Results (pages 17-18), and the table headings (new Table 4 and supplementary Tables S5a-d and S6a-c) to make this clearer. Because the analysis was originally multivariate, there has been no change to the analysis method. 

Comment 2:

In conclusion, based on Table 5, the authors mentioned that "management of hypertension and testing for proteinuria are important for successful sequential therapy that includes antiangiogenic drugs." However, there are no information about the real information about hypertension and proteinuria in the database. I think that it is impossible to produce this conclusion. I think that the checkup of proteinuria and blood pressure is essential for the treatment of CRC using ramucirumab. The doctor who do not check these parameters may not be able to manage any complications of chemotherapy, leading to bias.

Authors’ Response:

Thank you for raising this issue. We agree that the records of adverse event management procedures in the database do not necessarily indicate an occurrence of actual adverse events. We have emphasized this by adding the sentence below to the Discussion (page 23).

“Also, it is not clear whether patients actually experienced adverse events when these procedures for adverse event management were conducted.”

It is also true that our findings are observational only and that no causal relationship between the use of proteinuria tests or the presence of an antihypertensive drug prescription and continuation of therapy can be concluded. To address this, we have modified the words “important for“ to “associated with” throughout the manuscript to avoid claiming causal relationships. These changes have been made in the conclusion statement in the Abstract (page 4) and also in the Conclusions in the main text (page 24). We also added the statement below at the end of the Conclusions.

“As no causal relationship can be concluded from this retrospective study, further studies to evaluate causal relationships are warranted.”

Reviewer 2

Comment 1:

The manuscript is well written and shows a lot of real-world data of treatment sequence for CRC patients in Japan using a large administrative database. However, as authors mentioned in limitations session, the using database is not population based and may be far from general clinical data. For example, I recognized that the transition rate from 1st to 2nd line for CRC patients was almost 80%. On the other hand, this study showed the transition rate from 1st to 2nd line for presumed RAS-mutant CRC was 56.0% in Table 1. I think this MDV database lacks detailed clinical information.

Authors’ Response:

We thank the reviewer for raising this issue. While administrative healthcare databases are a valuable source of information on real-world treatment patterns and outcomes, the lack of clinical information and, in many cases, the inability to link the data to patient records, limits the interpretation of the findings. To address this, and in addition to the limitations already discussed in the Strengths and Limitations section of the Discussion (pages 24-25), we have added a new paragraph to the Introduction (page 6) that highlights the limitations of database research in broad terms, and we have expanded sections in the Discussion when interpreting specific results to ensure that the limitations to these interpretations are clear.

Comment 2:

Another point to mention, the authors suggest that management of hypertension and testing for proteinuria during treatment are important for successful continuation of sequential therapy that includes antiangiogenic drugs in conclusion session. Generally, patients who continue treatment for a long time have an increased risk of developing adverse events. So, it is difficult to say that management of hypertension and testing for proteinuria are important for successful continuation of treatment as conclusion. This conclusion could mislead the reader.

Authors’ Response:

We agree that our findings are observational only and that no causal relationship between the use of proteinuria tests or the presence of an antihypertensive drug prescription and continuation of therapy can be concluded. To address this, we have modified the words “important for“ to “associated with” throughout the manuscript to avoid claiming causal relationship. These changes have been made in the conclusion statement in the Abstract (page 4) and also in the Conclusions in the main text (page 24). We also added the statement below at the end of the Conclusions.

“As no causal relationship can be concluded from this retrospective study, further studies to evaluate causal relationships are warranted.”

Reviewer 3

Comment 1: Currently, one of the most relevant issues in CRC chemotherapy is the decision making of anti-VEGF drugs in second line treatment with FOLFIRI. However, this study is too much focus on Ramucirumab, and did not evaluate aflibercept or bevacizumab with FOLFIRI as second line chemotherapy. The authors should need to be re-evaluate all of issues in this study for these regimens, and please discuss the results. It is really helpful and clinically meaningful for clinicians.

Authors’ Response:

We thank the reviewer for this comment. To address this concern, we have revised the main analysis population from FOLFIRI plus ramucirumab to FOLFIRI plus any antiangiogenic drug available in Japan (ie, bevacizumab, ramucirumab, and aflibercept beta), to avoid focusing too much on a particular drug. The title of the manuscript has been modified to reflect this change and the main results now report findings for FOLFIRI plus any antiangiogenic drug. In addition, following your thoughtful recommendation to re-evaluate all of the issues in the study for all of the three regimens in order to provide information that is useful for clinical decision-making, we have added data for each antiangiogenic drug in the supplementary information (Supplementary Tables S2, S3a-c, S4, S6a-c). Table S4, which is new, summarizes the potential differences among patients who initiated each of the three regimens and provides information that may help inform clinicians’ decisions on which second-line treatment regimen to prescribe for their patients. While these changes have resulted in major edits to the Results and Discussion, there were no substantial differences in the interpretation of the results from the original manuscript. In addition, because we now report data across all three antiangiogenic drugs, we have taken this opportunity to include discussion about similarities and differences between the antiangiogenic drugs in the Results section and in the Discussion.

Comment 2: This manuscript is too much complicated, and it is very difficult to read and understand due to assessment of several aspects using complicated and not perfect MDV database. Therefore, it is better to re-write to focus more clinically relevant points to easy to understand for readers.

Authors’ Response:

We thank the reviewer for this feedback. In addition to changing the main analysis population from FOLFIRI plus ramucirumab to FOLFIRI plus any antiangiogenic drug, we have simplified the manuscript by deleting results that are less clinically relevant and which were not discussed in detail in the Discussion. We have removed the following from the manuscript:

- Original Table 2: distribution of ramucirumab prescriptions by treatment line

- Original Figure 3: overall treatment duration from third-line therapy

- Original Table 6: treatment regimen in the third line following second-line ramucirumab

- Original S2 Table: Concomitant procedures and medications received during second-line therapy in the combined first-line and early recurrence populations, by the second-line treatment

- Text only results that were on the original page 17 (corresponding to page 16 in the revised version), about overall treatment duration from start of second-line and from start of third-line, and transition rate from the second-line to third-line, for subgroups of patients who received ramucirumab only once vs two times or more during second-line therapy.

Although our revised manuscript now includes more supplemental tables than the original version, primarily because of the addition of subgroup analyses for the individual antiangiogenic drugs, we believe we have simplified the overall flow of the manuscript and have presented more clinically relevant information than in the previous version.

---

## [Decision Letter · Decision Letter 1]

17 Dec 2020

PONE-D-20-22549R1

Treatment sequences of patients with advanced colorectal cancer and use of second-line FOLFIRI with antiangiogenic drugs in Japan: A retrospective observational study using an administrative database

PLOS ONE

Dear Dr. Tanizawa Y,

Thank you for submitting your manuscript to PLOS ONE. After careful consideration, we feel that it has merit but does not fully meet PLOS ONE’s publication criteria as it currently stands. Therefore, we invite you to submit a revised version of the manuscript that addresses the points raised during the review process.

We look forward to receiving your revised manuscript.

Kind regards,

Takeshi Nagasaka, M.D., Ph. D.

Academic Editor

PLOS ONE

Additional Editor Comments (if provided):

Dear Authors,

As the revisers’ comments, this study is now ready to acceptance.

However, there still some concerns.

P20, Abstract, the sentence is difficult to follow, ‘Patients who underwent adjuvant therapy (and presumably experienced early recurrence) or first-line treatment for CRC in or after May 2016 were analysed until September 2019.’

This mean ‘this study enrolled CRC patients who underwent first-line chemotherapy, including the patients who received adjuvant therapy but presumably experienced early recurrence from May 2016 to September 2019’?

P27, Treatment sequences, the following sentences are very difficult to follow.

First-line or adjuvant therapy ended when patients added a new antitumor drug not included in the treatment regimen or when patients discontinued all antitumor drugs in the treatment regimen for 180 days or more, whichever occurred first.

Table 3.

Number of antiangiogenic drug prescriptions Median (IQR) 5 (3–9)

Please exclude mean (SD) 6.9 (6.1).

Reviewers' comments:

Reviewer's Responses to Questions

**Comments to the Author**

1. If the authors have adequately addressed your comments raised in a previous round of review and you feel that this manuscript is now acceptable for publication, you may indicate that here to bypass the “Comments to the Author” section, enter your conflict of interest statement in the “Confidential to Editor” section, and submit your "Accept" recommendation.

Reviewer #1: All comments have been addressed

Reviewer #2: (No Response)

2. Is the manuscript technically sound, and do the data support the conclusions?

Reviewer #1: Yes

Reviewer #2: Yes

3. Has the statistical analysis been performed appropriately and rigorously? 

Reviewer #1: Yes

Reviewer #2: Yes

4. Have the authors made all data underlying the findings in their manuscript fully available?

Reviewer #1: Yes

Reviewer #2: Yes

5. Is the manuscript presented in an intelligible fashion and written in standard English?

Reviewer #1: Yes

Reviewer #2: Yes

6. Review Comments to the Author

Reviewer #1: The authors described the treatment sequences for advanced colorectal cancer (CRC) and the use of ramucirumab in Japan, and to explore factors associated with overall treatment continuation with antitumor drugs.

The manuscript has been properly revised. I'm satisfied with this revised manuscript.

Reviewer #2: (No Response)

7. PLOS authors have the option to publish the peer review history of their article (what does this mean?). If published, this will include your full peer review and any attached files.

Reviewer #1: No

Reviewer #2: No

---

## [Author Response · Author response to Decision Letter 1]

6 Jan 2021

Editor

Comment 1: As the revisers’ comments, this study is now ready to acceptance. However, there still some concerns.

Authors’ Response: 

Thank you for your valuable feedback. We have explained below the changes made to the manuscript to address your concerns.

Comment 2: P20, Abstract, the sentence is difficult to follow, “Patients who underwent adjuvant therapy (and presumably experienced early recurrence) or first-line treatment for CRC in or after May 2016 were analysed until September 2019.” This mean ‘this study enrolled CRC patients who underwent first-line chemotherapy, including the patients who received adjuvant therapy but presumably experienced early recurrence from May 2016 to September 2019’?

Authors’ Response: The patient enrolment period was between May 2016 and July 2019, and the enrolled patients were followed up until September 2019. To clarify the sentence, we have modified the sentence to the following:

“Patients were enrolled if they started adjuvant therapy (and presumably experienced early recurrence) or first-line treatment for advanced CRC between May 2016 and July 2019, and were analysed until September 2019.”

Comment 3: P27, Treatment sequences, the following sentences are very difficult to follow. “First-line or adjuvant therapy ended when patients added a new antitumor drug not included in the treatment regimen or when patients discontinued all antitumor drugs in the treatment regimen for 180 days or more, whichever occurred first.”

Authors’ Response: As per your comment, we have amended the sentence to help clarify, as follows:

“The end of first-line or adjuvant therapy was defined either as 1) discontinuation of all antitumor drugs for 180 days or more, or 2) prescription of new antitumor drugs that were not included in the first-line or adjuvant treatment regimen, whichever occurred first.”

Comment 4: Table 3, Number of antiangiogenic drug prescriptions Median (IQR) 5 (3–9). Please exclude mean (SD) 6.9 (6.1).

Authors’ Response: Thank you for your suggestion. In Table 3, we have removed the mean (SD), and now show the median (IQR) only. Please note that we have made the same modification to S3a-c Tables.

---

## [Editor Report · Decision Letter 2]

15 Jan 2021

Treatment sequences of patients with advanced colorectal cancer and use of second-line FOLFIRI with antiangiogenic drugs in Japan: A retrospective observational study using an administrative database

PONE-D-20-22549R2

Dear Dr.Tanizawa,

We’re pleased to inform you that your manuscript has been judged scientifically suitable for publication and will be formally accepted for publication once it meets all outstanding technical requirements.

Kind regards,

Takeshi Nagasaka, M.D., Ph. D.

Academic Editor

PLOS ONE

Additional Editor Comments (optional):

Dear Authors,

You addressed all concerns.

This article is enough to giving acceptance.
---

## [Editor Report · Acceptance letter]

25 Jan 2021

PONE-D-20-22549R2 

Treatment sequences of patients with advanced colorectal cancer and use of second-line FOLFIRI with antiangiogenic drugs in Japan: A retrospective observational study using an administrative database 

Dear Dr. Tanizawa:

I'm pleased to inform you that your manuscript has been deemed suitable for publication in PLOS ONE. Congratulations! Your manuscript is now with our production department. 

Kind regards, 

on behalf of

Dr. Takeshi Nagasaka 

Academic Editor

PLOS ONE